# Age and Gender Trends in the Prevalence of Markers for Hepatitis E Virus Exposure in the Heterogeneous Bulgarian Population

**DOI:** 10.3390/life13061345

**Published:** 2023-06-08

**Authors:** Elitsa Golkocheva-Markova, Chiydem Ismailova, Ani Kevorkyan, Ralitsa Raycheva, Sashka Zhelyazkova, Stanislav Kotsev, Maria Pishmisheva, Vanya Rangelova, Asya Stoyanova, Viliana Yoncheva, Tencho Tenev, Teodora Gladnishka, Iva Trifonova, Iva Christova, Roumen Dimitrov, Roberto Bruni, Anna Rita Ciccaglione

**Affiliations:** 1NRL Hepatitis Viruses, Department of Virology, National Center of Infectious and Parasitic Diseases, 1504 Sofia, Bulgaria; chiydem.ismailova@abv.bg (C.I.); tdtenev60@gmail.com (T.T.); 2Department of Epidemiology and Disaster Medicine, Faculty of Public Health, Medical University, 4002 Plovdiv, Bulgaria; ani_kevorkian@mail.bg (A.K.);; 3Department of Social Medicine and Public Health, Faculty of Public Health, Medical University, 4002 Plovdiv, Bulgaria; dirdriem@gmail.com; 4Clinic of Nervous Diseases, University Hospital “Alexandrovska”, Medical University, 1431 Sofia, Bulgaria; doc_sashka@abv.bg; 5Department Infectious Diseases, Regional Hospital, 4400 Pazardzhik, Bulgaria; kotsevstanislav@gmail.com (S.K.); pishmishevampeleva@abv.bg (M.P.); 6NRL Enteroviruses, Department of Virology, National Center of Infectious and Parasitic Diseases, 1504 Sofia, Bulgaria; asq_ivo@abv.bg; 7NRL of Vector-Borne Infections, Listeria and Leptospires, Department of Microbiology, National Center of Infectious and Parasitic Diseases, 1504 Sofia, Bulgaria; teodorahristova@abv.bg (T.G.); iva_christova@yahoo.com (I.C.); 8Institute of Mathematics and Informatics, Bulgarian Academy of Sciences, 1000 Sofia, Bulgaria; roumendimitrov@gmail.com; 9Department of Infectious Diseases, Istituto Superiore di Sanità, 00161 Rome, Italy; roberto.bruni@iss.it (R.B.); annarita.ciccaglione@iss.it (A.R.C.)

**Keywords:** hepatitis E, prevalence, past infection, recent/ongoing infection, heterogeneous population

## Abstract

The prevalence of hepatitis E virus (HEV) in the Bulgarian population remains underestimated. The aim of the present study was to evaluate age and gender trends in HEV prevalence in the heterogeneous Bulgarian population. Stored serum samples from blood donors and different patient sub-populations—kidney recipients (KR), patients with Guillain–Barre syndrome (GBS), Lyme disease (LD), patients with liver involvement and a clinical diagnosis other than viral hepatitis A and E (non-AE), hemodialysis (HD) and HIV-positive patients (HIV)—were retrospectively investigated for markers of past and recent/ongoing HEV infection. The estimated overall seroprevalence of past infection was 10.6%, ranging from 5.9% to 24.5% for the sub-populations evaluated, while the seroprevalence of recent/ongoing HEV infection was 7.5%, ranging from 2.1% to 20.4%. The analysis of the individual sub-populations showed a different prevalence with respect to sex. In regard to age, the cohort effect was preserved, as a multimodal pattern was observed only for the GBS sub-population. Molecular analysis revealed HEV 3f and 3e. The type of the population is one of the main factors on which the anti-HEV prevalence depends, highlighting the need for the development of guidelines related to the detection and diagnosis of HEV infection with regard to specific patient populations.

## 1. Introduction

Hepatitis E virus (HEV) is a single-stranded, positive-sense RNA virus, whose virion exists in two forms—non-enveloped (spread between different hosts) and quasi-enveloped (circulating within the host organism) [1]. Based on genetic divergence, HEV is divided into eight genotypes and 31 sub-genotypes, which have been associated with geographic origin, host tropism, and clinical outcomes of HEV infection [2]. According to the current taxonomy, HEV refers to the family *Hepeviridae*, genus *Paslahepevirus*, as the members most commonly associated with infection in humans belong to the species *Paslahepevirus balayani* [3]. HEV infection has two distinct epidemiological patterns. The first is typical of highly endemic areas (Indian subcontinent, southeastern and central Asia, the Middle East, and northern and western parts of Africa) and is characterized by large outbreaks and a large proportion of cases with sporadic acute hepatitis. It is caused by HEV genotypes 1 and 2 and affects predominantly young healthy persons with a male predominance [4]. The infection is particularly severe in pregnant women and can progress to fulminant hepatitis in up to 30% [5]. The main mode of transmission is fecal-oral, mainly through contaminated water [6]. The second epidemiological pattern is typical for low endemic areas (Europe, United States, Canada, Australia, New Zealand, and several developed countries in Asia), with the predominant distribution of HEV genotypes 3 and 4, which are zoonotic. Humans are infected by the consumption of undercooked meat or by close contact with infected animals (swine, wild boar, and deer) or their products [6]. The infection predominantly affects middle-aged or elderly males and is characterized by a higher frequency of non-specific symptoms [7], such as neurological symptoms and disorders, hematological disorders, renal disease, acute pancreatitis, myocarditis, arthritis, and autoimmune thyroiditis [8]. In addition to animal reservoirs, HEV genotype 3 has been isolated from various environmental matrices, such as contaminated surface water [9], wastewater treatment plants [10], and even ticks [11]. Chronic HEV infection, which is typical for genotype 3 and rare for genotype 4, has been reported in immunocompromised patients, such as solid organ transplant patients, HIV-infected, or patients with hematologic malignancies [12].

The differential diagnosis of HEV infection is based on the detection of specific antibodies or HEV RNA in serum. Usually, acute HEV infection is confirmed by the presence of anti-HEV IgM, which remain detectable at the onset of symptoms. However, there is increasing evidence of the long-lasting persistence of this antibody class [13,14]. Recent infection can also be confirmed by the simultaneous detection of anti-HEV IgM and anti-HEV IgG or the rising titers of the last one in the serum [15]. Anti-HEV IgG is a marker of past or chronic infection and persists for a long time—years or decades, with a decreased titer after a period of 5 years and is characterized by a low rate of seroreversion [16]. HEV RNA is detected in serum during the early phase of acute infection. Its presence in serum is a marker of ongoing HEV infection. Detection of HEV RNA is mainly applied in the diagnosis of immunocompromised patients. In the case of HEV reinfection, detectable HEV RNA in combination with rising anti-HEV IgG levels in the absence of anti-HEV IgM is a marker of recent infection [17].

The established global anti-HEV IgG seroprevalence in the general population was 12.47%, for anti-HEV IgM was 1.47%, and for HEV RNA was 0.20%. In special populations, the measured anti-HEV IgG seroprevalence was 16.91% in HIV-positive patients, 13.10% in hemodialysis patients, and 11.68% in organ transplant recipients [18]. For Europe, anti-HEV IgG seroprevalence in the general population varied between 1.7% and 34%, as the heterogeneity was established not only between but also within countries [19]. Human cases were mainly caused by locally acquired HEV-3 strains [20]. For the indigenous HEV infection, the predominant affection of males compared to females [21,22] and the increasing prevalence of anti-HEV IgG with age [23] have been established. Among blood donors, HEV seroprevalence ranged from 4.7% to 29.5% [24]. In Bulgaria, established anti-HEV IgG prevalence varied from 9.04% to 28.8%, with the circulation of HEV genotype 3 [25].

The aims of this retrospective study were (1) to evaluate gender and age trends in the prevalence of anti-HEV IgG, as a marker of past HEV infection, and anti-HEV IgM, as a marker of recent/ongoing HEV infection, in 773 individuals distributed in 7 sub-populations according to clinical condition or characteristics (samples collected in Bulgaria in 2020–2022) and (2) to analyze HEV genotypes/sub-genotypes within the HEV RNA positive samples.

## 2. Materials and Methods

### 2.1. Population and Study Design

Retrospective samples of interest from patients and blood donors were recruited from stored blood serum samples at the NRL Hepatitis Viruses for the period of three years 2020–2022. Samples were collected as part of public health and institutional surveillance activities; ethical approval was not required. Serologically, only anti-HEV seropositivity was evaluated for the project aims, taking into account samples with (1) past HEV infection, i.e., anti-HEV IgG (+) and anti-HEV IgM (−) and HEV RNA (−) (anti-HEV IgG alone), and (2) recent or ongoing HEV infection, i.e., anti-HEV IgM (+) and/or anti-HEV IgG (+) and/or HEV RNA (+). Demographic data (age and sex) were taken into account. Overall, the study population consisted of 773 subjects distributed among 7 specific target sub-populations: (1) blood donors (BD) *n* = 94; (2) kidney recipients (KR) *n* = 23; (3) patients with Guillain–Barre syndrome (GBS) *n* = 49; (4) patients with Lyme disease (LD) *n* = 94; (5) patients with liver involvement and a clinical diagnosis other than viral hepatitis A to E—chronic liver diseases, malignancies, or alcohol abuse (non-AE) *n* = 56; (6) patients on hemodialysis (HD) *n* = 218; and (7) HIV-positive patients (HIV) *n* = 239. With the exception of blood donors, the other sub-populations included in the study were characterized by increased HEV burden, i.e., due to increased risk of zoonotic exposure (for LD), development of extrahepatic neurological symptoms (for GBS), or increased risk of chronic HEV infection (for KR, HD, non-AE, HIV).

### 2.2. Virological Assays

The serum samples were tested for anti-HEV IgG and IgM using diagnostic ELISA (Euroimmun, Lübeck, Germany or DiaPro, Sesto San Giovanni, Italy) according to the manufacturer’s instructions. The tested samples were considered anti-HEV IgM reactive at a signal/cut-off ratio (S/CO) of ≥1.1 for Euroimmun and ≥1.2 for DiaPro assay. For anti-HEV IgG, the results were interpreted semi-quantitatively. The samples were considered anti-HEV IgG positive in a S/CO ratio of ≥1.1 for both assays. IgM-positive samples were further tested for HEV RNA, and positive ones were genotyped.

Detection and quantification of HEV RNA were performed by RealStar HEV RT-PCR kit2.0 (Altona diagnostics, Hamburg, Germany) in accordance with the manufacturer’s instructions. The test runs were considered valid at the generated standard curve control parameter (R^2^) of ≥0.98. The determined manufacturer linear range of the RT-PCR was from 1 × 10^1^ to 1 × 10^7^ IU/µL.

### 2.3. Genotyping/Subgenotyping

HEV genotyping was performed by sequencing the ORF2 region of the HEV genome according to the method of Boxman et al. [26]. HEV RNA was automatically extracted on ExiPrep 16Dx (Bioneer, RK). Reverse transcription was performed with a specific HEV gene primer using SuperScript III First-Strand Synthesis System for RT-PCR (Invitrogen, Waltham, MA, USA), in accordance with the manufacturer’s instructions. Nested PCR was carried out with primer sequences made available by RIVM to European laboratories participating in the HEVnet database. For the first PCR step, the forward primer was from positions 5909–5934 of the HEV genome sequence (Accession Number M73218), and the reverse primer was complementary to positions 6512–6534. For the nested PCR, the forward and reverse primers were complementary, respectively, to positions 5948–5985 and 6479–6513. The amplification products were analyzed by agarose gel electrophoresis. After purification, PCR products were sequenced using the GenomeLab Dye Terminator Cycle Sequencing (DTCS) Quick Start Kit on an automated DNA sequencer (Beckman Coulter, Inc., Fullerton, CA, USA) in accordance with the manufacturer’s instructions. Initially obtained isolates were genotyped on Hepatitis E Virus Genotyping Tool Version 0.1 (RIVM). The output sequences were aligned and consensus sequences were generated in BioEdit. HEV genotype/subtype was assigned by phylogenetic analysis of a dataset containing the recent case HEV sequences (*n* = 2, GenBank accession numbers OQ613225 and OQ613224), previously assigned sequences from Bulgarian patients with acute HEV infection (*n* = 10, GenBank accession numbers MH203175, MH203178, MH203181, MH203185, MH203198, MH203200, MH203201, MH203210, MH203222, MH203223) [27], and the recommended set of standard reference HEV sequences for different genotypes/subtypes [2]. The phylogenetic analysis was performed using the Neighbor-Joining method on the Molecular Evolutionary Genetics Analysis software version 11 (MEGA 11.0).

### 2.4. Statistical Analysis

Depending on the sample distribution, continuous data were expressed as mean ± SD or median with interquartile range (IQR). HEV exposure frequencies (anti-HEV IgG, anti-HEV IgM, and concomitant detection of anti-HEV IgG and IgM) were reported for the overall population and individual sub-populations based on selected covariates—gender and seventh different age groups (decades). Numbers and percentages (*n*, %) were used to present qualitative variables. The z-test was applied to assess differences between proportions of interest. The effect of the predictor variable on the outcome variable was determined by linear regression modelling, and R2 estimated the model’s quality and explained the observed variance. A 2-sided *p*-value of <0.05 was considered statistically significant. Statistical analyses were performed using SPSS Statistics v. 26 software (IBM Corp., Chicago, IL, USA).

## 3. Results

### 3.1. Main Characteristics of the Study Population/Sub-Populations

The median age of the overall population (773 samples) was 46 years (25th; 75th—34; 62), as the youngest (8 years old) was from the HIV sub-population and the oldest (93 years old) were from non-AE and HD sub-populations (Table 1). The highest percentage of samples, 23.5%, were distributed in the 30–39 age group, with the under-30 years’ age group being the least represented (12.4%). Among the seven sub-populations, the most represented age groups were: 30–39 for blood donors and HIV-positive patients, 40–49 for kidney recipients and patients with Guillain–Barre syndrome, 50–59 for patients with Lyme disease, and older than 70 years for patients with non-AE and for hemodialysis patients.

Of all enrolled 773 subjects, 66.4% were male and 33.6% were females (Table 1), with females predominating in patients with Lyme disease and with a non-AE disease, respectively, male to a female ratio of 1:1.1 and 1:1.4. The males represented more than 50% in the sub-populations of blood donors, kidney recipients, patients with GBS, hemodialysis, and HIV-positive patients.

Of all 773 samples, 82/773 (10.6%) were positive for past HEV infection (anti-HEV IgG(+)/IgM(−)) and 58/773 (7.5%) were positive in regard of recent/ongoing HEV infection (anti-HEV IgM(+)/IgG(−) or IgG(+)) (Table 1). The highest prevalence of serological markers for past (24.5%) and recent/ongoing (20.4%) HEV infection was established for the GBS sub-population. A high prevalence of past infection was also found in the non-AE (17.8%), BD (15.9%), and LD (12.7%) sub-populations, and of recent HEV infection in the KR (17.4%) and the BD (14.9%) sub-populations. The lowest rate for past infection was found in the HD sub-population (5.9%), and for recent/ongoing HEV infection in the LD sub-population (2.1%). Two samples were HEV RNA positive—one from the KR, and one from the GBS sub-population.

### 3.2. Gender Trend in the Seropositivity for Past and Recent/Ongoing HEV Infection

To investigate the gender trend in HEV prevalence, the samples were stratified by males and females. In general, men and women were almost equally affected by past HEV infection, respectively, 10.9% (56/513) and 10.0% (26/260) (Figure 1). For the recent/ongoing HEV infection, men exceeded women almost twice, respectively, 9.0% (46/513) and 4.6% (12/260) (z = 2.2, *p*-value = 0.028). The trend for a higher prevalence of past infection among males persisted in the BD (18.2%), KR (15.4%), GBS (29.0%), LD (13.6%), and HD (6.5%) sub-populations in comparison with females, respectively, 5.9%, 0%, 16.7%, 12.0%, and 5.3%. In the sub-populations of the non-AE and HIV-positive patients, the dependence was impaired and the prevalence was higher among women (21.2% and 10.5%) compared to men (13.0% and 7.0%). Such a reverse trend was established for the prevalence of recent/ongoing HEV infection for the BD sub-population, where the females were more affected in comparison with males, respectively, 17.6% vs. 14.3%. A statistically significant higher percentage of recent/ongoing HEV-infected GBS patients were males (29.0%, *n* = 9) compared to females (5.6%, *n* = 1) (z = 2.0, *p*-value = 0.046). Serological markers for recent HEV infection were not detected among the women from the LD sub-population.

### 3.3. Age Trend in the Seropositivity for Past and Recent/Ongoing HEV Infection

To evaluate HEV antibodies prevalence by age the studied population was stratified by seven age decades: <30; 30–39; 40–49; 50–59; 60–69; ≥70, and for sub-population analysis due to the small number of samples the KR sub-population was merged with the HD one. When analyzing the age trend within the overall population (Figure 2) the linear regression model demonstrates that for the past HEV infection, age variance explained 35.4% of the variance in positive cases. Each decade increase in age adds 0.8% to the number of positive cases. A statistically significant higher percentage of past HEV-infected individuals were over 70 years old (15.0%, *n* = 16) compared to persons in their 30s (10.4%, *n* = 10) (z = 2.2, *p*-value = 0.031). Regarding the recent/ongoing HEV infection, age variance explained 61.0% of the variance in positive cases. Each increase in age decade resulted in a 1.1% decrease in positive cases.

When comparing the age decades within individual sub-populations, the prevalence of anti-HEV IgG followed three distinct patterns (Figure 3). For the BD, nAE, HD, and HIV sub-populations, a stepwise increase in prevalence was found with advancing age. Some variations were detected, such as the absence of anti-HEV IgG up to age 50 within the non-AE sub-population, for age group < 30 years within the hemodialysis patients, and for the 50–59 age decade for HIV-positive patients as well as fluctuations in the trend, up to 2.9% for the 40–49 age group in the HD and to 4.0% for the age group 30–39 within the HIV sub-population. An inversely dependent pattern was observed for the LD sub-population, where the anti-HEV IgG seroprevalence decreased from 50% to 4.5% with age. As for the GBS sub-population, the multi-modal age distribution with the highest age incidence of 36.4% in the 50–59 age group, followed by 33.3% in <30 and ≥70 age groups were measured.

In contrast, the anti-HEV IgM seroprevalence, as a marker for recent/ongoing infection, was characterized by an inverse correlation with age for most of the sub-populations, as the absence of anti-HEV IgM in individual age groups was noted (Figure 3). This age trend was broken in the LD sub-population, where the markers for recent/ongoing HEV infection (8.3%) were detected only in the 50–59 age group.

### 3.4. Detection of HEV RNA, Genotyping, and Phylogenetic Analysis

Two patients tested positive for HEV RNA—one from the KR, and the other from the GBS sub-population. The viral load was, respectively, 94,188 [IU/mL] and 4800 [IU/mL]. Both patients were males, respectively, 59 and 36 years old. The isolates were genotyped as HEV 3. The sub-clustering assigned the sequences as subtype 3e for kidney recipient patient and subtype 3f for patient with GBS. The phylogenetic analysis revealed the close relation of both sequences with other isolates from Bulgarian patients with acute viral hepatitis E (Figure 4).

## 4. Discussion

This retrospective study focused on evaluating the prevalence of HEV in Bulgaria using serum samples from a heterogeneous population consisting of different patient groups and blood donors. Gender and age trends in HEV seroprevalence for overall and sub-populations were analyzed. The overall seroprevalence was 10.6% for the past and 7.5% for recent/ongoing HEV infection and males were found to be more affected than females by a recent/ongoing infection. For the overall population, the percent of samples positive for past HEV infection increased with age, while for recent/ongoing infection an inverse correlation was found, with the frequency of positive samples decreasing with increasing age. The analysis of the individual sub-populations showed a different predominance with respect to gender, with more women affected in the sub-populations of blood donors, patients with underlying chronic liver disease, and HIV-positive patients. With regard to age, the typical cohort effect for HEV prevalence was observed in most sub-populations with the exception of the GBS and the LD sub-populations.

### 4.1. Anti-HEV Prevalence

Among sub-populations, the prevalence ranged from 5.9% to 24.5% for past infections and from 2.1% to 20.4% for recent HEV infections.

Patients with clinically diagnosed Guillain–Barre syndrome were most affected by HEV, with a prevalence of 24.5% and 20.4% for past and recent infections, respectively, and one viremic case (HEV 3f). High anti-HEV IgG seroprevalence was reported in German patients with GBS (41%); the rate of recent HEV infection, which the authors defined as anti-HEV IgM(+)/IgG(+)/RNA(−) status, was 1.2% [28].

HEV seroprevalence higher than the overall was established also for the BD sub-population. In contrast, another study conducted between 2020 and 2021 found an extremely higher prevalence among blood donors, with an anti-HEV IgG prevalence of 25.9%, with geographic heterogeneity ranging from 21.3% to 28.8% [29]. This discrepancy may result from the different approaches used to differentiate the past and recent/ongoing HEV infection. In fact, in the present study, some anti-HEV IgG positive samples were excluded from computation of “past infections” because they were also positive for anti-HEV IgM and, so, were classified as “recent/ongoing infections”.

Patients with underlying liver disease, with different malignancies, or increased alcohol consumption, are at risk of developing post-HEV infection complications. A study from England showed that hepatitis E infection was the cause of acute severe liver injury in 5% of patients with underlying liver disease [30]. Another study from the USA detected anti-HEV IgG in 16% of adult patients with solid tumors and hematologic malignancies [31], which may explain the high burden of past HEV infection observed in the non-AE sub-population (17.8%) in the present study.

HEV and the Lyme disease spirochete *Borrelia burgdorferi* [32] are zoonotic pathogens with occupational exposure as a risk factor for human infection. Although it does not require a transmissible vector such as *B. burgdorferi*, the isolation of HEV from ticks, collected from Eurasian wild boar, has been reported [11]. Univariate analysis of HEV positivity among forest workers in France showed an association with positivity for Lyme disease [33], which is consistent with the increased prevalence of past HEV infection in the LD sub-population (12.7%) observed in the present study.

HEV infection is often associated with chronicity, the development of hepatocellular carcinoma in kidney transplants [34], or allograft rejection [35]. The pooled HEV seroprevalence in kidney recipient patients worldwide has been estimated to be 15.3% [36], with the highest value of 40% reported in French recipients [37]. The seroprevalence, defined in this study (17.4% for recent HEV infection), is in line with the global one, which defines the need for a more accurate diagnosis of HEV in solid organ transplant recipients.

### 4.2. Trends in the Prevalence of Anti-HEV by Gender

The role of sexual dimorphism in infectious diseases, especially in viral infections, is often underestimated. For example, HBV and HCV infections are more common in males [38]. With regard to the anti-HEV IgG prevalence, it is assumed that males and females are equally affected [39], which was confirmed in the present study in the context of the overall population, although males are generally over-represented demographically both in the overall population and in individual sub-populations. An earlier study on a Bulgarian heterogeneous cohort with a similar overall prevalence of 9.04% for past HEV infection (10.6% in this study) also found no differences between males and females [40]. In contrast, in the sub-populations of blood donors, kidney recipients, patients with Guillain–Barre syndrome, and patients with Lyme disease, the number of men, positive for past HEV infection, exceeded that of women. In fact, a systematic review found that in the Americas (with documented HEV genotypes 1 and 3) young immunocompetent females (6–39 years) have higher HEV seropositivity compared to males, but in the case of immunosuppression, being male was a risk factor for HEV [41]. Thus, the underlying immunosuppression in kidney recipients, to prevent an allograft loss [42] and in patients with Lyme disease, as a result of the established immunosuppressive effect of borrelia infection [43], can explain the predominance of males. It is worth mentioning that neurological complications were not rare in acute HEV infection. Data from France revealed that neurological symptoms were developed in 22% of acute immunocompetent HEV cases [44]. Considering the suggestion that the HEV genotype 3 infection (which is common in European countries) predominantly affects elderly men compared to women [45], this can explain the male predominance in the GBS sub-population in the present study. In contrast, in the non-AE and HIV sub-populations, the females were characterized by a slightly higher prevalence of past HEV infection. It is well known that females are characterized by an increased prevalence of acute liver failure [46]. At the same time, progressive liver fibrosis caused by past HEV genotype 3 infection has been described in female patients [47], which is in accordance with the increased prevalence of past HEV infection in non-AE women in our study. The established gender trend for the HIV sub-population differs from our earlier work on this group [48]. The reason is that in the present study, we have used different analytical approaches in the context of screening to evaluate the prevalence of HEV markers. Although the detection of HEV antibodies is a reliable diagnostic approach, a precise diagnostic algorithm for laboratory confirmation of HEV infection has not yet been specified. Anti-HEV IgM is considered the first differential marker, which in combination with the rising anti-HEV IgG or the presence of HEV RNA is a marker for acute HEV infection [17]. The DiaPro ELISA assay used in the current study, for the detection of anti-HEV IgM, is characterized by the ability to remain positive for the longest time after the onset of the HEV infection compared to other immuno-enzyme assays [49]. For this reason, in context of screening, the presence of anti-HEV IgG alone was used as a marker for past HEV infection, and anti-HEV IgM in combination with IgG or HEV RNA or their absence was used as a marker for recent/ongoing HEV infection.

### 4.3. Trends in the Prevalence of Anti-HEV by Age

Age related seroprevalence helps to understand the age-specific susceptibility to an infectious disease. The age-cohort effect has been demonstrated for HEV prevalence in many studies, where a statistically significant higher seroprevalence was detected for participants over 50 years of age compared to the 20–30 age group [50]. In the present study, such dependence was found for past HEV infection, where HEV-infected individuals over 70 years of age exceeded those in the 30–39 age group. The same trend with some discrepancies was followed in the BD sub-population, non-AE, HD, and HIV sub-populations. In contrast, the reverse trend was detected for recent/ongoing HEV infection, with prevalence decreasing with older age. In general, less than 2% of individuals infected with HEV are symptomatic [51]. At the same time, the Belgian study revealed that patients infected with HEV subtype 3f were at a high risk of hospitalization compared with those infected with subtype 3c, and the median age of hospitalized patients was 59 years [52]. Considering that subtypes 3f and 3e accounted cumulatively for 85% of HEV genotype 3 cases in Bulgaria [27], the decline in the number of retrospectively screened samples positive for markers of recent/ongoing infection with increasing age can be explained.

The older age (mean age 56 years) of HEV positivity was also demonstrated in German outpatients with chronic liver disease also [53]. Regarding the HD sub-population, the results were similar to other studies in hemodialysis patients where an increase in anti-HEV IgG was detected in patients over 40 years of age [54]. In contrast, disruption in the age trend was observed in the GBS and LD sub-populations. The German study of GBS patients found an age-dependent anti-HEV IgG prevalence with a peak in the 60–69 age group [28]. In our study, the pattern of past HEV prevalence was multimodal with high age incidence in the <30, 50–59, and ≥70 age groups. A bi-modal age pattern was typical for the prevalence of the GBS, with the highest age incidence in the 5–9 and 60–64 age groups and a rapid decline in the oldest age [55]. Thus, the established multimodal age pattern might be the result of underestimation of HEV circulation in Bulgaria or the possible neurogenic potential of circulating strains. For the LD sub-population, the age trend was inversely proportional—the younger age groups were characterized by increased anti-HEV IgG seroprevalence. It is known that loss of anti-HEV IgG can occur in 30% of naturally infected immunocompetent patients who have cleared the infection [56], which together with the already mentioned immunosuppressive effect of Borrelia infection [43] may explain the decreased anti-HEV IgG with increasing age. In fact, in the LD sub-population, markers of recent/ongoing HEV infection were detected only in the 50–59 age group, suggesting impaired immune response in patients with Lyme disease.

### 4.4. HEV3 Subtypes

HEV genotype 3 is the most prevalent in Europe, with the majority of strains belonging to subtypes 3c, 3f, and 3e [57]. HEV genotype 3 has been reported as a causative agent of glomerular manifestations in solid organ transplant recipients [34] and also as the triggering agent of neurological symptoms [58]. The genotyping of the sequences, obtained from KR and GBS patients, revealed subtypes 3f (KR patient) and 3e (GBS patient). Despite the perception that neurological injuries are not HEV subtype-specific [58], the neurotropic potential in our study was demonstrated for subtype 3e. Our findings correlated with the study of Bruni et al., where the distribution of HEV genotype 3 among Bulgarian patients with acute viral hepatitis E was detected, with HEV 3f and 3e being the predominant subtypes [27].

### 4.5. Limitations

The present study has several limitations. The main one is that this is a retrospective cohort study with a limited number of patients in some of the groups, which may lead to a lack of uniformity and limit the possibility of finding more in depth significance. Due to the small number of patients in the KR sub-population, we were not able to analyze the age trend for it. During the study period, the two types of immune-enzyme assays were used, which did not allow us to look for an exact correlation with other studies.

## 5. Conclusions

To the best of our knowledge, this is the first study to simultaneously analyze the gender and age trends in different Bulgarian sub-populations. Previous studies have analyzed HEV prevalence in strictly defined populations, such as patients with acute viral hepatitis [59,60], hemodialysis patients [61], and blood donors [29]. The type of the study population is one of the three main factors, together with the serological test used and the geographical region, on which the anti-HEV prevalence depends and which each accounted, respectively, for 7% heterogeneity in seroprevalence [19]. The applied analysis clearly distinguished differences in the HEV prevalence pattern in different patient groups according to their sex and age, indicating that guidelines related to the detection and diagnosis of HEV infection should be closely tailored to the type of population.

## Figures and Tables

**Figure 1 life-13-01345-f001:**
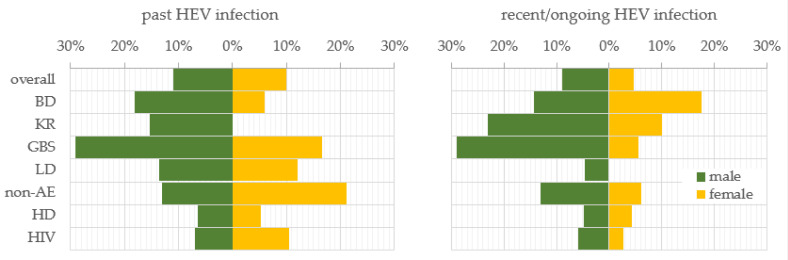
Gender trend in the prevalence of serological markers for past and recent/ongoing HEV infection. HEV prevalence was expressed as a percentage of positive samples out of the total tested samples for both genders—male and female, for the overall population and, respectively, sub-populations. Abbreviations: overall = overall population; BD = blood donors; KR = kidney recipients; GBS = patients with Guillain–Barre syndrome; LD = patients with Lyme disease; non-AE = patients with liver involvement and a clinical diagnosis other than viral hepatitis A to E—chronic liver diseases, malignancies, or alcohol abuse; HD = patients on hemodialysis; HIV = HIV-positive patients.

**Figure 2 life-13-01345-f002:**
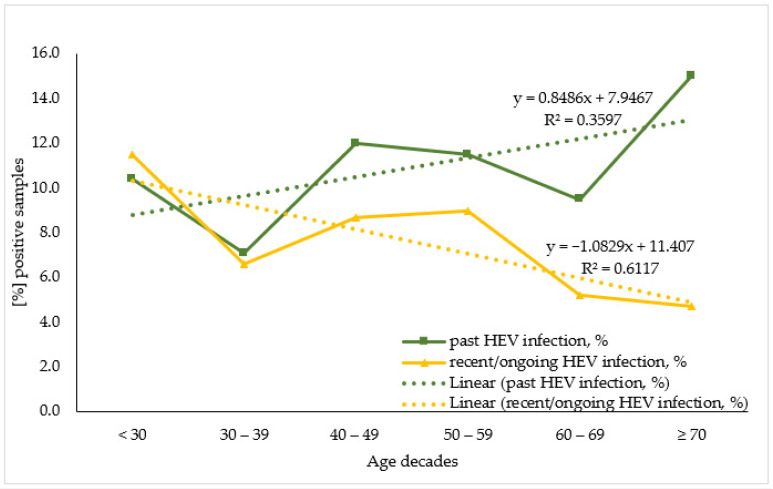
Age trend in seroprevalence of the markers for past and recent/ongoing HEV infection in the overall population.

**Figure 3 life-13-01345-f003:**
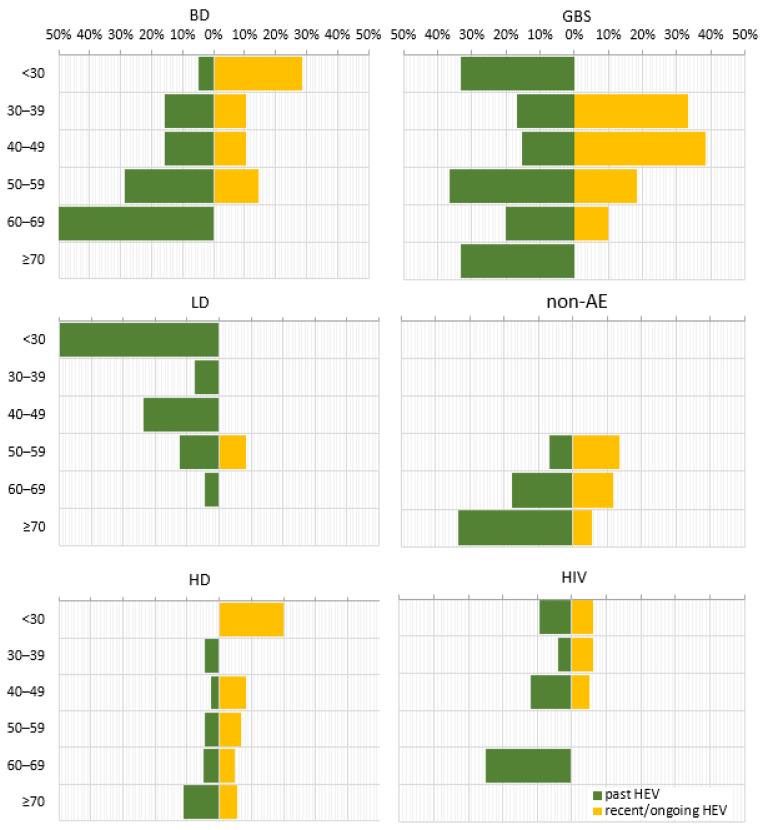
Age trend in the prevalence of serological markers for past and recent/ongoing HEV infection for different sub-populations. HEV prevalence was expressed as a percentage of positive samples out of the total tested samples for different age groups. Abbreviations: overall = overall population; BD = blood donors; KR = kidney recipients; GBS = patients with Guillain–Barre syndrome; LD = patients with Lyme disease; non-AE = patients with liver involvement and a clinical diagnosis other than viral hepatitis A to E—chronic liver diseases, malignancies, or alcohol abuse; HD = patients on hemodialysis; HIV = HIV-positive patients. Age decades: <30; 30–39; 40–49; 50–59; 60–69; ≥70.

**Figure 4 life-13-01345-f004:**
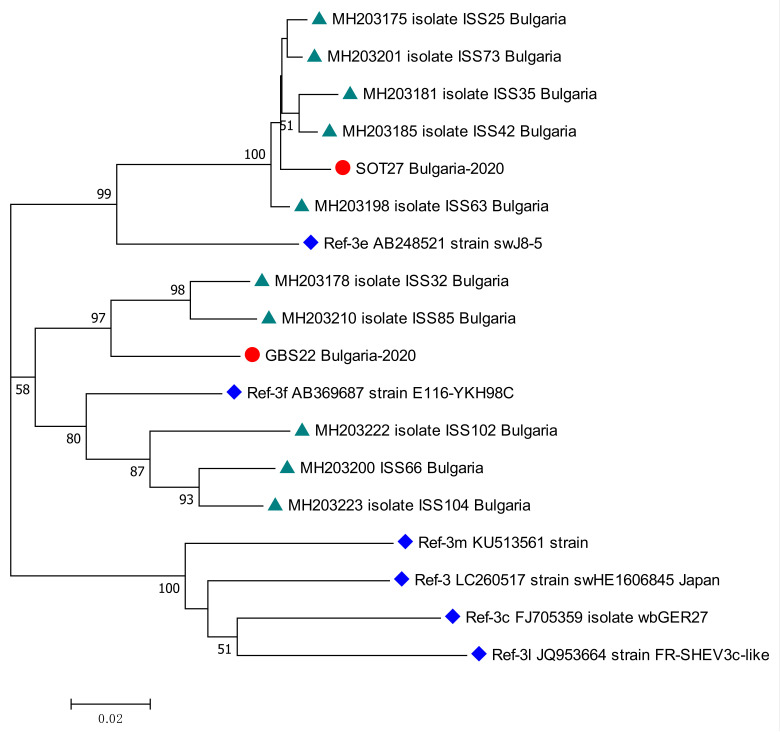
Phylogenetic tree with the 2 HEV case sequences from a kidney recipient and from a patient with GBS. Red circle represents sequences from KR (SOT27 Bulgaria-2020, GenBank accession number OQ613225) and GBS (GBS22 Bulgaria-2020, GenBank accession number OQ613224) cases; green triangle—HEV sequences isolated from Bulgarian patients with acute viral hepatitis E infection (MH203175, MH203178, MH203181, MH203185, MH203198, MH203200, MH203201, MH203210, MH203222, MH203223) [27]; blue rhombus—reference strains for HEV genotype 3 (LC260517) subtypes 3e (AB248521), 3f (AB369687), 3m (KU513561), 3c (FJ705359) and 3l (JQ953664) [2]. The bootstrap values > 50 were reported.

**Table 1 life-13-01345-t001:** The main characteristics of evaluated variables of the studied overall population and separated sub-populations.

Characteristic	BD [*n* = 94]	KR [*n* = 23]	GBS [*n* = 49]	LD [*n* = 94]	non-AE [*n* = 56]	HD [*n* = 218]	HIV [*n* = 239]	Overall Population [*n* = 773]
Age [years]: median	36	45	52	55	64	63	34	46
25th; 75th percentiles	30; 44	36; 49	42; 63	42; 62	55; 71	52; 71	29; 42	34; 62
min–max	19–60	24–71	27–83	17–78	34–91	24–93	8–69	8–93
Age decades *n* (%):								
<30	21 (22.3)	2 (8.7)	3 (6.1)	4 (4.3)	-	3 (1.4)	63 (26.4)	96 (12.4)
30–39	38 (40.4)	5 (21.7)	6 (12.2)	13 (13.8)	3 (5.4)	17 (7.8)	100 (41.8)	182 (23.5)
40–49	19 (20.2)	11 (47.8)	13 (26.5)	21 (22.3)	3 (5.4)	25 (11.5)	58 (24.3)	150 (19.4)
50–59	14 (14.9)	4 (17.4)	11 (22.4)	24 (25.5)	15 (26.8)	40 (18.3)	14 (5.9)	122 (15.8)
60–69	2 (2.1)	-	10 (20.4)	22 (23.4)	17 (30.4)	61 (28.0)	4 (1.7)	116 (15.0)
≥70	-	1 (4.3)	6 (12.2)	10 (10.6)	18 (32.1)	72 (33.0)	-	107 (13.8)
Gender *n* (%): male	77 (81.9)	13 (56.5)	31 (63.3)	44 (46.8)	23 (41.1)	124 (56.9)	201 (84.1)	513 (66.4)
female	17 (18.1)	10 (43.5)	18 (36.7)	50 (53.2)	33 (58.9)	94 (43.1)	38 (15.9)	260 (33.6)
m:f	4.5:1	1.3:1	1.7:1	1:1.1	1:1.4	1.3:1	5.3:1	2:1
HEV prevalence ^1^: past infection *n* (%) IgG(+)/IgM(−)	15 (15.9)	2 (8.7)	12 (24.5)	12 (12.7)	10 (17.8)	13 (5.9)	18 (7.5)	82 (10.6)
recent/ongoing infection n (%): IgM(+)/IgG(−) or IgG(+)	14 (14.9)	4 (17.4)	10 (20.4)	2 (2.1)	5 (8.9)	10 (4.6)	13 (5.4)	58 (7.5)
HEV RNA *n* (%):	-	1 (4.3)	1 (2.0)	-	-	-	-	2 (0.3)

^1^ HEV prevalence was expressed as a percentage of positive samples out of the total tested samples for the population/sub-population. Abbreviations: BD = blood donors; KR = kidney recipients; GBS = patients with Guillain–Barre syndrome; LD = patients with Lyme disease; non-AE = patients with liver involvement and a clinical diagnosis other than viral hepatitis A to E—chronic liver diseases, malignancies, or alcohol abuse; HD = patients on hemodialysis; HIV = HIV-positive patients; IgM = immunoglobulin M; IgG = immunoglobulin G.

## Data Availability

The corresponding author can provide the data that was presented in this study upon request.

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
