# Peer review of "Age and Gender Trends in the Prevalence of Markers for Hepatitis E Virus Exposure in the Heterogeneous Bulgarian Population"

_life, 2023, doi:10.3390/life13061345_

Round 1

Reviewer 1 Report

Interesting and important study

--------------------

-What is the main question addressed by the research?

Prevalence of HEP E in general population in Bulgaria

-Do you consider the topic original or relevant in the field? Does it address a specific gap in the field?

Yes it relevant and original The research is on general population and not only on specific cohorts

-What does it add to the subject area compared with other published material?

The research is on general population and not only on specific cohorts

-What specific improvements should the authors consider regarding the methodology? What further controls should be considered?

The methodology is well established and acceptable

The patients sample  can be bigger

-Are the conclusions consistent with the evidence and arguments presented and do they address the main question posed?

yes

-Are the references appropriate?

YES

-Please include any additional comments on the tables and figures.

No

Author Response

Response to Reviewer 1 Comments

Interesting and important study

Point 1: - What specific improvements should the authors consider regarding the methodology? What further controls should be considered?

The methodology is well established and acceptable

The patients sample can be bigger

Response 1: Thank you for this comment. We appreciate the reviewer’s comments regarding the number of patient samples. Indeed, when dividing the individuals into different populations, the number of samples in some of them (in the kidney transplant recipients and in the patients with Guillain-Barre syndrome) turned out to be small (< 50). However, these two sub-populations were included in the design of the analysis because they are characterized by some of the highest levels of the serological markers tested. At the same time, these are some of the most important patient groups in which hepatitis E virus infection can lead to long-term sequelae or atypical symptoms. Therefore, in selecting relevant patient groups, we felt that their inclusion would be helpful in tracking trends by sex and age in the prevalence of markers of recent/ongoing and past hepatitis E virus infection. At the same time, our study is retrospective and includes samples from the NRL Hepatitis Virus Serum Bank that were submitted for analysis during the three-year period 2020-2022 (page 3, lines 103-104). In addition, the samples were part of the Laboratory’s public health and institutional surveillance activities (page 3, lines 109-110). For clarity, and at the reviewer's suggestion, we have made an edit and combined the two sentences into a single paragraph at the beginning of the Study Design subsection: “Retrospective samples of interest from patients and blood donors were recruited from stored blood serum samples at the NRL Hepatitis Viruses for the period of three years 2020-2022. Samples were collected as part of public health and institutional surveillance activities; ethical approval was not required.”

Reviewer 2 Report

Markova et al. conducted a study to investigate the occurrence of hepatitis E virus (HEV) in a specific sub-population in Bulgaria, focusing on age and gender as key factors. The researchers examined stored serum samples from various donors and patients with different diseases to assess both past and current prevalence rates. Their analysis revealed that age in decades, gender (male vs. female), past infections, and recent/ongoing infections were significant determinants of HEV prevalence. Moreover, the authors observed distinct gender-related trends in past and current HEV prevalence among donors and patients with different diseases. Notably, they found variations in the prevalence trends across different age groups when comparing past and recent rates of HEV infection.

The manuscript provides a comprehensive overview of the prevalence of hepatitis E virus (HEV) among patients with various diseases and donors within the Bulgarian population. The authors have conducted a thorough analysis and presented the findings in a highly detailed manner. The manuscript contains comprehensive information derived from the analysis, and no significant comments or suggestions arise from the evaluation.

Minor comments:

1.      To enhance clarity in the discussion section, it is advisable to incorporate subheadings wherever appropriate. These subheadings can highlight specific topics such as "trend with age", "limitations," allowing for a more organized and focused discussion of the findings.

2.      For easy reference, it is recommended to include abbreviations for terms like BD, KR, LD, etc., under Table 1. By providing these abbreviations, readers can conveniently refer back to the corresponding terms within the table.

Minor correction is required

Author Response

Response to Reviewer 2 Comments

The manuscript provides a comprehensive overview of the prevalence of hepatitis E virus (HEV) among patients with various diseases and donors within the Bulgarian population. The authors have conducted a thorough analysis and presented the findings in a highly detailed manner. The manuscript contains comprehensive information derived from the analysis, and no significant comments or suggestions arise from the evaluation.

Minor comments:

Point 1. To enhance clarity in the discussion section, it is advisable to incorporate subheadings wherever appropriate. These subheadings can highlight specific topics such as "trend with age", "limitations," allowing for a more organized and focused discussion of the findings.

Response 1: Thank you for this comment. We have added the following subheading in the Discussion section (pages 9 to 11) in response to Your suggestion: “4.1. Anti-HEV prevalence (line 301); 4.2. Trends in the prevalence of anti-HEV by gender (line 337); 4.3. Trends in the prevalence of anti-HEV by age (line 378); 4.4. HEV3 subtypes (line 413); 4.5. Limitations (line 424)”

Point 2. For easy reference, it is recommended to include abbreviations for terms like BD, KR, LD, etc., under Table 1. By providing these abbreviations, readers can conveniently refer back to the corresponding terms within the table.

Response 2: This was amended as recommended. On page 5, in the legend of Table 1, all abbreviations used are explained: “Abbreviations: overall = overall population; BD = blood donors; KR = kidney recipients; GBS = patients with Guillain-Barre syndrome; LD = patients with Lyme disease; non-AE = patients with liver involvement and a clinical diagnosis other than viral hepatitis A to E - chronic liver diseases, malignancies, or alcohol abuse; HD = patients on hemodialysis; HIV = HIV-positive patients. Age decades: < 30; 30-39; 40-49; 50-59; 60-69; >=70”. Explanations are also inserted for Figure 1 (page 6) and Figure 3 (page 8)

Language editing of the manuscript was also done as per the reviewer's recommendations.